# Rare Pediatric Cerebellar High-Grade Gliomas Mimic Medulloblastomas Histologically and Transcriptomically and Show p53 Mutations

**DOI:** 10.3390/cancers16010232

**Published:** 2024-01-04

**Authors:** Zhi-Feng Shi, Kay Ka-Wai Li, Anthony Pak-Yin Liu, Nellie Yuk-Fei Chung, Chit Chow, Hong Chen, Nim-Chi Amanda Kan, Xian-Lun Zhu, Danny Tat-Ming Chan, Ying Mao, Ho-Keung Ng

**Affiliations:** 1Department of Neurosurgery, Huashan Hospital, Fudan University, Shanghai 200040, China; shizhifeng@fudan.edu.cn; 2Hong Kong and Shanghai Brain Consortium (HSBC), Hong Kong, China; 3Department of Anatomical and Cellular Pathology, The Chinese University of Hong Kong, Hong Kong, China; kayli@cuhk.edu.hk (K.K.-W.L.); yf169chung@cuhk.edu.hk (N.Y.-F.C.); chit@cuhk.edu.hk (C.C.); 4Department of Paediatrics and Adolescent Medicine, The University of Hong Kong, Hong Kong, China; apyliu@hku.hk; 5Department of Paediatrics and Adolescent Medicine, Hong Kong Children’s Hospital, Hong Kong, China; 6Department of Pathology, Huashan Hospital, Fudan University, Shanghai 200040, China; cherrychen30@126.com; 7Department of Pathology, Hong Kong Children’s Hospital, Hong Kong, China; amandakan@yahoo.com; 8Division of Neurosurgery, Department of Surgery, The Chinese University of Hong Kong, Hong Kong, China; zhuxl@surgery.cuhk.edu.hk (X.-L.Z.); tmdanny@surgery.cuhk.edu.hk (D.T.-M.C.)

**Keywords:** children, high-grade gliomas, p53, PDGFRA, MYCN

## Abstract

**Simple Summary:**

High-grade gliomas (HGGs) in the cerebellum of children have been rarely described. We studied the histological and molecular features of a series of five pediatric high-grade gliomas in the cerebellum. These unique cases showed histological and immunohistochemical similarities to medulloblastoma, which is a main differential diagnosis of poorly differentiated tumors in the cerebellum in children. Furthermore, these tumors showed high scores by NanoString-based transcriptomic assay for medulloblastoma. Genomic methylation profiling, however, revealed that they clustered to the glioblastoma subclasses. TP53 mutations were found in all cases by panel sequencing. This study adds to the rare pathological and molecular characterization of pediatric cerebellar high-grade gliomas and shows that their histological, immunohistochemical, and transcriptomical characteristics overlap with those of medulloblastoma. We recommend the use of both methylation array and TP53 screening in the differential diagnoses of poorly differentiated embryonal-like tumors of the cerebellum.

**Abstract:**

Pediatric high-grade gliomas (HGG) of the cerebellum are rare, and only a few cases have been documented in detail in the literature. A major differential diagnosis for poorly differentiated tumors in the cerebellum in children is medulloblastoma. In this study, we described the histological and molecular features of a series of five pediatric high-grade gliomas of the cerebellum. They actually showed histological and immunohistochemical features that overlapped with those of medulloblastomas and achieved high scores in NanoString-based medulloblastoma diagnostic assay. Methylation profiling demonstrated these tumors were heterogeneous epigenetically, clustering to GBM_MID, DMG_K27, and GBM_RTKIII methylation classes. MYCN amplification was present in one case, and PDGFRA amplification in another two cases. Interestingly, target sequencing showed that all tumors carried TP53 mutations. Our results highlight that pediatric high-grade gliomas of the cerebellum can mimic medulloblastomas at histological and transcriptomic levels. Our report adds to the rare number of cases in the literature of cerebellar HGGs in children. We recommend the use of both methylation array and TP53 screening in the differential diagnoses of poorly differentiated embryonal-like tumors of the cerebellum.

## 1. Introduction

Brain tumors are the most common solid tumors in children and the leading cause of childhood solid cancer deaths in children [1]. Around half of the CNS tumors in children are gliomas [2]. In children, low-grade gliomas account for the majority of cases, whereas high-grade gliomas (HGG) are infrequent [3,4]. High-grade gliomas in children in the cerebellum are very rare, accounting for approximately <1% of all pediatric brain tumors [3,4,5].

Many large series of pediatric high-grade gliomas contain no or very few cerebellar cases, and detailed molecular features have not been described or separated from those of supratentorial pediatric HGGs [6,7,8]. In a series of 51 pediatric high-grade gliomas, only a single case was located in the cerebellum [9]. In another large meta-analysis of 1069 high-grade gliomas in children, there were only 21 cerebellar-midline cases [10]. Only 5 of the 21 cases were examined for methylation profiling and mutations [10]. Reinhardt et al. described a large series of 86 cerebellar glioblastomas of different ages [11]. Twenty-five cases of this series were by methylation profiling high-grade astrocytomas of piloid features, generally a circumscribed tumor found in adults [12,13]. Only thirteen cases of this large series were pediatric [11]. Five of them were also secondary to previous radiation treatment [11]. Mutational profiling was available only in 5/13 cases, and three of the cases carried IDH mutations. These patients were, however, 15 years or above and thus belonged to young adult gliomas [11]. In a recent study, Buccoliero et al. described 11 pediatric high-grade gliomas, and only one case was located in the cerebellum [14]. Histological anaplasia in pilocytic astrocytomas has been described, but this is currently only categorized as a subtype of pilocytic astrocytoma by the World Health Organization (WHO) Classification (2021), and they may be confused with high-grade astrocytomas of piloid features mentioned above [13,15].

World Health Organization Classification (2021) now classifies many pediatric-type high-grade gliomas into an IDH-wildtype H3-wildtype group. Korshunov et al. [16] classified such tumors into pedGBM_RTK1 (enriched for PDGFRA amplification), pedGBM_RTKII (enriched for EGFR amplification), and pedGBM_MYCN (enriched for MYCN amplification) groups, but only 4/87 cases of their series were cerebellar. So, overall, cerebellar HGGs in children are poorly documented and characterized.

Although there is insufficient information concerning rare cerebellar high-grade gliomas in children, a key differential diagnosis in poorly differentiated cerebellar tumors in children is medulloblastoma, as both are histologically poorly differentiated neuroepithelial tumors. In this study, we found from the archives of our two institutions five cases of pediatric cerebellar high-grade gliomas that not only histologically closely mimicked medulloblastomas but also transcriptomically overlapped with the latter. We further studied them by methylation profiling, bulk RNAseq, and target sequencing. They clustered to glioma-related subtypes by methylation profiling and showed copy number variations (CNVs) found in glioblastomas. We further demonstrated that all cases harbored TP53 mutations.

## 2. Materials and Methods

### 2.1. Patients

The cases were identified from the archives of the Chinese University of Hong Kong and Huashan Hospital, Fudan University, Shanghai. We reviewed cases from our hospital systems cerebellar high-grade gliomas and medulloblastomas for the period 2002–2022 for the Chinese University of Hong Kong and 2013–2022 for Huashan Hospital, Shanghai. The latter only routinely admitted pediatric patients at a later period. The cases described in this report, including three cases previously diagnosed as medulloblastomas as described in the Results, were the only cases of cerebellar high-grade gliomas in children we could eventually identify. In the same period, there were 192 pediatric medulloblastomas resected at our institutions. Histological review was carried out by two neuropathologists (HKN and HC). Histological examination and immunohistochemistry were as per routine practice.

### 2.2. Fluorescence In Situ Hybridization (FISH)

BRAF fusion, MYCN amplification, and PDGFRA amplification were evaluated by FISH. BAC clone (RP11-355H10) containing the genomic sequences of chromosome 2p24 and the centromeric probe (CEP 2 (D2Z1), Vysis) were employed to detect MYCN alteration. For PDGFRA amplification, PDGFRA probes (CTD-2054G11 and RP11-231C18) and centromeric probes (CEP 4, Vysis) were used. For KIAA1549-BRAF fusion, 3 clones of P1-derived artificial chromosomes spanning the entire BRAF gene (RP4-726N20, RP5-839B19, and RP4-813F11) and a centromeric enumeration probe for chromosome 7 (CEP7) were used. The target probes were labeled with Spectrum Orange, and reference probes were labeled with Spectrum Green. FISH criteria were described in our previous studies [17,18]. At least 100 non-overlapping signals were scored and analyzed in this study. BRAF fusion was considered when BRAF/CEP7 ratio was ≥1.15, and more than 20% of tumor cells showed relative BRAF gain [19]. MYCN and PDGFRA amplifications were defined when >5% of cells displayed clusters or a ratio of target to reference signal > 2 [17].

### 2.3. Methylation Profiling

Genome-wide methylation profiling was performed on FFPE tissues by Sinotech Genomics Co., Ltd., Shanghai, China, and as previously used by us [20]. Briefly, DNA was extracted, bisulfite modified, restored, and hybridized to an Illumina Infinium Methylation EPIC 850K BeadChip array. After hybridization, arrays were washed and scanned. Signal intensities in the IDAT files were subjected to background correction and dye-bias normalization, according to Capper et al. [21]. Sex chromosome probes and probes targeting the known SNPs were excluded from the analysis. T-SNE (t-distributed stochastic neighbor embedding) plot was generated with Rtsne R package v0.13 according to Capper et al. [21]. IDAT files were also uploaded to DKFZ molecular classifier (www.molecularneuropathology.org, accessed on 1 January 2024).

### 2.4. Identification of Copy Number Variations with 850K Array

Detection of copy number variations (CNV) was conducted similar to our previous publication [22]. Briefly, the ‘conumee’ R package in Bioconductor was used to determine copy number variations. The cutoff for amplification/loss and homozygous loss were log2 ratios ±0.35 and −0.415, respectively [23].

### 2.5. Target Sequencing

Target sequencing was performed on FFPE tissues, as previously used [20]. Briefly, DNA was extracted and purified from FFPE tissues with truXTRAC FFPE Kits (Covaris, Woburn, MA, USA). Libraries were prepared using KAPA HyperPrep kit (Roche, Cape Town, South Africa). The DNA libraries were evaluated for quality and quantity prior to library sequencing with HiSeq platform (Illumina, San Diego, CA, USA). Paired-end reads were aligned to the hg19 (GRCh37) build of the human reference genome. Variants were called and annotated with smCounter2 and wANNOVAR. Variants that did not pass the quality filters, had variant allele fractions < 10%, had variant allele counts ≤ 5, or had minor allele frequencies > 1% in 1000 Genomes or gnomAD exome databases were excluded from further analysis.

### 2.6. Detection of TERT Promoter Mutation

TERT promoter mutations were detected by Sanger sequencing as per our previous study [24,25]. In brief, crude cell lysate was prepared from FFPE sections. DNA from the lysate was mixed with forward primer, reverse primer, and KAPA HiFi HotStart ReadyMix (Roche, Cape Town, South Africa) in a PCR reaction. The primer sequences were TERT-F (5′-GTCCTGCCCCTTCACCTT-3′) and TERT-R (5′-CAGCGCTGCCTGAAACTC-3′), and the amplified fragment spanned the two mutational hotspots (C228T and C250T) in the TERT promoter region. PCR products were visualized on electrophoresis gel, purified, and sequenced with BigDye Terminator Cycle Sequencing kit v1.1 (Life Technologies, Carlsbad, CA, USA).

### 2.7. Target RNA Sequencing

For target RNA sequencing, total RNA was extracted with RNeasy FFPE kit (Qiagen, Hilden, Germany). The quantity and quality were evaluated, and RNA passing the quality check was converted into cDNA. Libraries were then prepared with TruSight RNA Pan-Cancer Target Genes (Illumina), which was designed to examine 1385 cancer-related genes. Paired-end reads were aligned to human genome assembly GRCh37 (hg19). STAR aligner and STAR fusion caller were employed to call for fusion genes.

### 2.8. Nanostring-Based Affiliation

A Nanostring-based assay was used previously to detect the transcript abundance of 22 medulloblastoma group-specific genes and 3 housekeeping genes [26]. Together with an R script, the assay provides molecular subgroup prediction for medulloblastoma by a confidence score. This assay was widely used in the molecular subgrouping of medulloblastomas [26,27]. In brief, RNA samples from FFPE tissues were extracted using RNeasy FFPE Kit (Qiagen) and then were assessed by NanoDrop 2000 spectrophotometer. RNA samples were then hybridized to the NanoString nCounter CodeSet at 67 °C for 20 h. Hybridized samples were then purified and immobilized on cartridges with the nCounter Prep Station (NanoString Technologies, Seattle, WA, USA). Fluorescent signals were then read by the nCounter Digital Analyzer (NanoString Technologies). Raw data was then normalized with R package ‘NanoStringNorm’, and predictions for medulloblastoma subgroups were made with package ‘pamr’.

### 2.9. Immunohistochemistry

GFAP, Ki67, Olig2, Synaptophysin, and p53 were examined by immunohistochemistry. Briefly, FFPE tissues of 4 µm thickness were dewaxed in xylene and rehydrated in graded alcohol. Immunohistochemical staining was performed in BenchMark ULTRA automated tissue staining system (Ventana Medical Systems, Rotkreuz, Switzerland) or Bond Max automated platform (Leica Biosystems, Nußloch, Germany). Sections were incubated with the GFAP (Dako, Z0334, 1:2500), Ki67 (Thermo Fisher, Waltham, MA, USA, SP6, 1:200), Olig2 (IBL-America, Minneapolis, MN, USA 18953, 1:120), Synaptophysin (Novocastra, Newcastle Upon Tyne, UK, NCL-L-SYNAP-299, 1:50), and p53 (Dako, Santa Clara, CA, USA, DO-7, 1:40). Immunostaining was detected with the OptiView DAB IHC Detection Kit (Ventana Medical Systems). A tumor was scored as p53 positive if >10% of tumor nuclei showed strong nuclear staining [28].

## 3. Results

The clinical features of this series of five patients are presented in Appendix A. Four cases had very poor overall survival (OS) ranging from five months to a year, and four patients received adjuvant therapy post-surgery (Appendix A). The one surviving patient was a recent case. The MR images and histological features are shown in Figure 1 and Appendix A. The radiologic images of one case, however, were already archived by the hospital and could not be found. By imaging, all five cases were somewhat centrally rather than peripherally located in the cerebellum. The epicenters of these tumors were all located in the cerebellar hemisphere or peduncles.

Histologically, all five cases closely resembled medulloblastomas with structureless sheets of closely packed, monomorphic, embryonal-like blue cells with hyperchromatic nuclei and little cytoplasm. Homer-Wright rosettes were absent. Morphological features of astrocytes-like fibrillary cells or gemistocytes were also absent. No nodules were seen. No feature of pilocytic astrocytoma was present. Small foci of necrosis (cases 3 and 5) and microvascular proliferation (cases 3 and 4) were noted. Immunohistochemically, all tumors stained strongly for synaptophysin (one case focal) and also strongly for Olig2 (one case focal), but only variably for GFAP. Ki67 labeling was very high for all five tumors. They also stained strongly for p53 (one case focal) (Figure 1 and Appendix A).

Two cases (#1 and 2) were interrogated by the Nanostring-based assay for molecular grouping of medulloblastomas at the time of diagnosis. High confidence scores were obtained, and these cases were assigned medulloblastoma molecular subgroups (Figure 2). Two other cases were now similarly successfully studied, and one case achieved a high score (Figure 2). One older case was unsuccessfully studied by Nanostring. Three cases, including the two cases with initial high confidence scores, were diagnosed as medulloblastomas with the corresponding molecular subgroups at the initial diagnoses (cases 1, 2, and 5). The other two cases were diagnosed with high-grade gliomas at initial diagnoses.

Methylation profiling using Illumina Infinium 850K chips was performed retrospectively for all cases, and they were mapped against the tSNE map of the DKFZ Molecular Neuropathology reference cohort. According to Capper et al., glioblastomas can be epigenetically divided into GBM RTK II and GBM mesenchymal (MES), and less frequently, GBM RTK I, GBM RTK III, GBM MID, GBM MYCN, and GBM G34 [21,29]. Three cases were clustered in proximity to or overlaid with the GBM_MID class. One case was clustered in proximity to the DMG_K27 class, and one other case was clustered to the GBM_RTKIII class (Figure 3). Copy number variation (CNV) analysis of methylation profiling identified MYCN amplification in one case and PDGFRA amplification in another two cases (Figure 2). These amplifications were confirmed by fluorescence in situ hybridization (FISH) (Figure 1 and Appendix A). As mentioned in the Introduction, PDGFRA and MYCN amplifications were characteristic of some pediatric H3-wildtype high-grade gliomas [16].

We further carried out a custom-designed panel DNA sequencing on all five cases (Appendix A). TP53 mutations were found in all five tumors, four being missense and one being frame-shift mutation consistent with the immunostaining (Figure 1 and Figure 2, Appendix A). Other recurrent mutations identified in this study were KMT2D and PDGFRA mutations. H3F3A, HIST1H3B, BRAF, TERT promoter, and IDH mutations were not detected in all cases. The absence of H3 mutations was important in spite of their somewhat central locations in the posterior fossa, as that excluded infiltration of the cerebellum from a diffuse infantile pontine glioma (DIPG), which is usually H3K27-mutated. Germline mutations were screened by targeted sequencing in two patients where blood samples were available, and one patient showed PBRM1 mutation (p.D163N) and CCND1 mutation (p.A270V). Both variants were regarded as of unknown significance by standard guidelines. No germline TP53 mutations could be identified.

RNA sequencing was conducted to test for the presence of fusion genes. Only three cases (cases 1, 3, and 5) showed sufficient materials for the assay, and no fusion was identified. All cases were also tested for BRAF fusion by FISH, and they were all negative.

## 4. Discussion

Pediatric cerebellar high-grade gliomas are rare, and our current understanding of the molecular alterations underlying these tumors is scarce. Only a very small number of cerebellar tumors have been included in the large cohorts of pediatric high-grade gliomas as mentioned in the Introduction. Our small series shows that rare pediatric cerebellar high-grade gliomas may display histological, immunohistochemical, and transcriptomic similarities to medulloblastomas. All five tumors of this series histologically resembled medulloblastomas with sheets of closely packed hyperchromatic embryonal-like cells with scanty cytoplasm. Moreover, histologically, they all showed strong immunostaining for synaptophysin, which is one of the commonest stains used for the diagnosis of medulloblastoma in pathology practice [30]. Apart from ETMR, the mimics of medulloblastoma do not usually express neuronal markers such as synaptophysin [30]. Conversely, Olig2 or GFAP positivity should only be found in a small number of cells in medulloblastomas [30,31,32]. In three of our cases, necrosis or microvascular proliferation was present; these histologic features are not unique to glioblastomas but can be used as clues in such diagnostic scenarios in addition to Olig2.

The nanostring-based assay has been widely used in the diagnostic and molecular grouping of medulloblastomas [27], but it is also known that a small number of non-medulloblastoma brain tumors may be misdiagnosed by the assay. Of 239 cases of “medulloblastomas” evaluated by Nanostring by Korshunov et al. [33], 18 cases (8%) were finally diagnosed as other brain tumors by methylation profiling. The three cases illustrated in this series show that cerebellar high-grade gliomas can achieve very high confidence scores in Nanostring assay for medulloblastoma molecular groups, in addition to histological similarity to medulloblastomas, and support the notion that methylation profiling should be used as the investigation of choice for studying poorly differentiated cerebellar tumors in children.

All five tumors showed TP53 mutations, which could be found in nearly half of the non-brainstem pediatric high-grade gliomas and in almost all H3F3A G34V/R gliomas [7]. As detailed in the Introduction, cerebellar HGGs are rare, and only a few have been molecularly characterized. Zhang et al. examined 854 pediatric brain tumors of different types by whole genome sequencing (WGS) and/or RNAseq and identified only four cerebellar high glioma gliomas, and three of them had TP53 mutations. The tumor without TP53 mutations carried NF1 mutation [34]. Also, the only pediatric cerebellar high-grade gliomas in the Buccoliero et al. study showed TP53 mutation, and this was the sole mutation found in this tumor [14]. Furthermore, in a previous study of 32 radiation-induced pediatric gliomas, 2 out of the 8 radiation-induced, cerebellum high-grade gliomas carried TP53 mutations [35]. Interestingly, TP53 is also mutated in a subgroup of SHH-medulloblastomas with worsened progress [36]. Immunostaining for p53 is a useful screening method for TP53 mutation in routine pathology practice [37].

In summary, we provided detailed histological and genomic characterization for a small series of rare sporadic pediatric cerebellar high-grade gliomas. They may mimic medulloblastoma histologically, immunohistochemically, and transcriptomically. Our findings support the notion that in addition to the usual tools to differentiate embryonal tumors, methylation profiling should be used as the investigation of choice for studying poorly differentiated cerebellar tumors in children.

## 5. Conclusions

This study expands our knowledge of rare pediatric cerebellar high-grade gliomas. We showed that pediatric cerebellar high-grade gliomas can be transcriptomically and histologically similar to medulloblastomas. We recommend using methylation profiling whenever feasible, Olig2 and p53 staining in delineating differential diagnoses of poorly differentiated embryonal-like tumors of the cerebellum, in addition to the usual tools to differentiate other embryonal tumors like ATRT (atypical teratoid/rhabdoid tumor) and ETMR (embryonal tumor with multilayered rosettes).

## Figures and Tables

**Figure 1 cancers-16-00232-f001:**
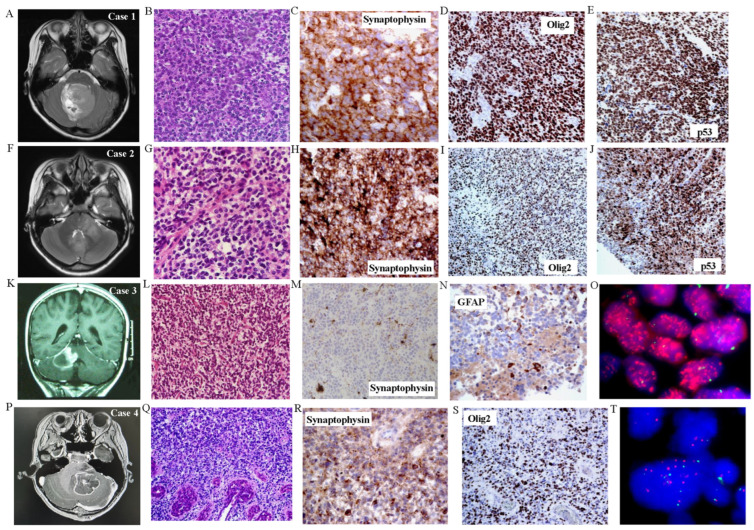
Histological and immunohistochemical features of tumor samples. (**A**–**E**) Case 1. (**A**) T2 MRI showing large fourth ventricular tumor. (**B**) H&E showing monomorphic sheets of small hyperchromatic embryonal-looking cells with no cytoplasm (×200). Tumor cells were positive for (**C**) synaptophysin (×400), (**D**) Olig2 (×200), and (**E**) p53 (×200). Tumor cells were negative for GFAP and NeuN, positive for INI-1, and showed high Ki67 labeling (not shown). (**F**–**J**) Case 2. (**F**) T2 MRI showing large tumor involving cerebellar peduncle and fourth ventricle. (**G**) H&E showing features similar to case 1 with sheets of primitive-looking cells (×200). Immunostaining for (**H**) synaptophysin (×400), (**I**) Olig2 (×200), and (**J**) p53 (×200). Tumor cells were negative for NeuN and focally positive for GFAP, positive for INI-1, and showed high Ki67 labeling (not shown). (**K**–**O**) Case 3. (**K**) T2 MRI shows tumor at cerebellar peduncle. Histology shows (**L**) groups of hyperchromatic cells with no cytoplasm packed together (×200) with small foci of necrosis and microvascular proliferation (not shown). Immunostaining for (**N**) synaptophysin (×200) and (**M**) GFAP (×400). Tumor cells were focally positive for Olig2, negative for NeuN, positive for INI-1, positive for p53, and showed high labeling for Ki67 (not shown). (**O**) FISH demonstrates PDGFRA amplification consistent with CNV findings. (**P**–**T**) Case 4. (**P**) T2 MRI showing large cerebellar tumor. (**Q**) H&E shows sheets of hyperchromatic embryonal cells with small foci of microvascular proliferation (×400). Expression of (**R**) synaptophysin (×400) and (**S**) Olig2 (×200). Tumor cells were focally positive for p53 and GFAP, negative for NeuN, positive for INI-1, and showed high labeling for Ki67. (**T**) FISH shows MYCN gains consistent with CNV findings.

**Figure 2 cancers-16-00232-f002:**
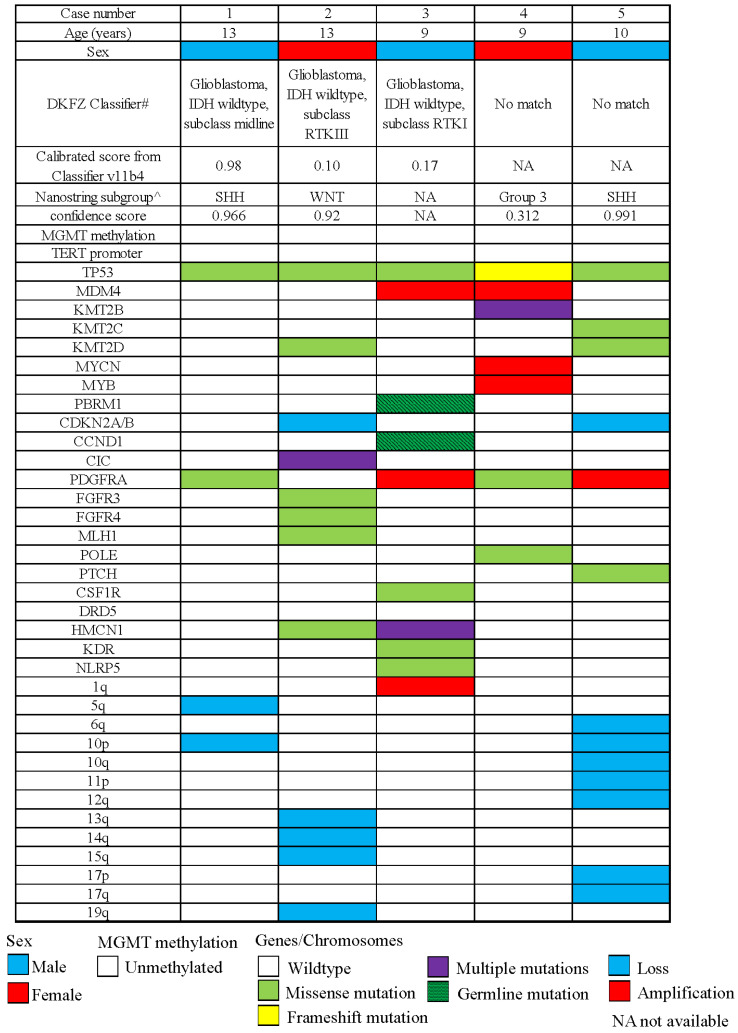
Oncoprint summary of clinical and molecular profiles of this cohort. ^ Nanostring transcriptome array as was done according to previous studies [26,27]. # Methylation class was assigned by DKFZ [21].

**Figure 3 cancers-16-00232-f003:**
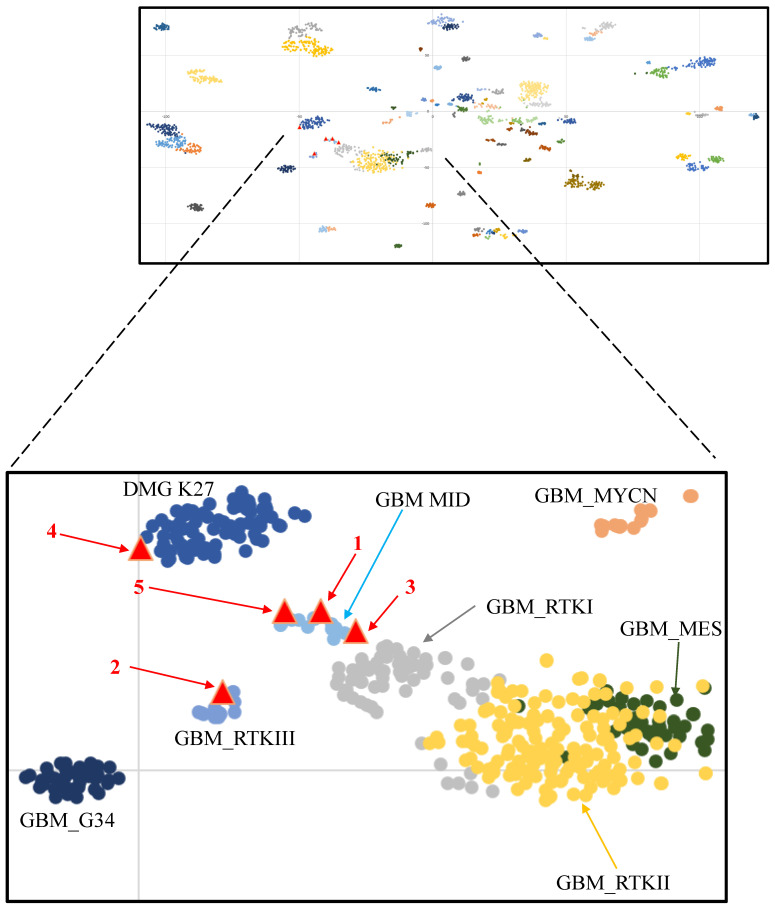
tSNE clustering of DNA methylation profiles of the five tumors (red triangles) alongside 2801 reference samples from Capper et al. [21].

## Data Availability

The IDAT files for this study can be found at https://www.surgery.cuhk.edu.hk/BTC/HSBC/ (accessed on 1 January 2024).

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
