# Peer review of "Rare Pediatric Cerebellar High-Grade Gliomas Mimic Medulloblastomas Histologically and Transcriptomically and Show p53 Mutations"

_cancers, 2024, doi:10.3390/cancers16010232_

Round 1

Reviewer 1 Report

Comments and Suggestions for Authors

The manuscript "Rare pediatric cerebellar high-grade gliomas mimic medullo-2 blastomas histologically and transcriptomically and show p53 3 mutations" by Zhi-Feng Shi and colleagues is  retrospective review of high grade cerebellar glioma cases from the archives of the Chinese University of Hong Kong and Huashan Hospital, Fudan University, Shanghai. Classification of these cases as high grade gliomas instead of medulloblastoma is only uniformly supported by the methylation studies while other methods are not conclusive. Moreover some of the cases like case 1 looks very suggestive for medulloblastoma.

1) Please provide the period considered by the revision of the archives and, if possible, the total number of pediatric gliomas and medulloblastomas treated in those years in the same Hospitals.

2) An MRI image of the lesion is provided only for 4 of the 5 patients described, please justify the lack of one image or add at least one image for all patients.

3) Why the MRI image of Case 4 is relegated to the supplemental materials and not included in figure 1?

3) Why the MRI image of Case 3 show a coronal section and all other images provided are axial? Either provide axial and coronal images for all cases or provide axial or sagittal images for all cases. Uniformity of representation make easier for the reader to compare the aspect of the different lesions.

4) The MRI images provided illustrating Cases 2 and 4 show a clear involvement of the pons. How authors can be sure that the high grade glioma in those patients did not originate in the pons (a rather common site for high grade gliomas in children) secondarily invading the cerebellum? Please discuss this possibility and provide indication that involvement of the pons was not present in the remaining cases.

Author Response

Comment:

The manuscript "Rare pediatric cerebellar high-grade gliomas mimic medullo-2 blastomas histologically and transcriptomically and show p53 3 mutations" by Zhi-Feng Shi and colleagues is  retrospective review of high grade cerebellar glioma cases from the archives of the Chinese University of Hong Kong and Huashan Hospital, Fudan University, Shanghai. Classification of these cases as high grade gliomas instead of medulloblastoma is only uniformly supported by the methylation studies while other methods are not conclusive. Moreover some of the cases like case 1 looks very suggestive for medulloblastoma.

Response:

Thanks for the helpful review.

Question 1:  Please provide the period considered by the revision of the archives and, if possible, the total number of pediatric gliomas and medulloblastomas treated in those years in the same Hospitals.

Response:  We retrieved cases from our archives for the periods 2002-2022 at the Chinese University of Hong Kong and 2013-2022 at Huashan Hospital, Shanghai.  The reason for the discrepancy was that Huashan Hospital, Shanghai only routinely admitted pediatric patients at a later period.  We retrieved from our system cerebellar high-grade gliomas and medulloblastomas to review and looked for potential cases, but did not look at all pediatric gliomas, as medulloblastoma is not a differential diagnosis for  supratentorial lesions.  The cases described, including the three cases previously diagnosed as medulloblastomas, were the only cases of cerebellar high-grade gliomas we could identify eventually.     In the same period, there were 192 pediatric medulloblastomas excluding adult cases and consult cases at our institutions.  We outlined these under subsection “2.1 Patients” of “Materials and Methods.”

Question 2:   An MRI image of the lesion is provided only for 4 of the 5 patients described, please justify the lack of one image or add at least one image for all patients.

Response:       The MR images of one patient (case 5) were removed already from the hospital information system.  We only had the textual radiology report.  The textual radiology reports of all five cases were available.

Question 3:   Why the MRI image of Case 4 is relegated to the supplemental materials and not included in figure 1?

Response:       The illustrations for the other three cases are already quite numerous and if the photos concerning case 4 were included, it would have made Figure 1 too crowded.   Also, as explained under point 2, this case was a bit not like the other cases, as we could not find the MR images so we put it under Supplementary.  Hope that this is allowed by the Reviewer.

Question 4:   Why the MRI image of Case 3 show a coronal section and all other images provided are axial? Either provide axial and coronal images for all cases or provide axial or sagittal images for all cases. Uniformity of representation make easier for the reader to compare the aspect of the different lesions.

Response:       Similarly to point 3, the MR images of case 3 were actually removed from the hospital information system.   Fortunately, one of the co-authors had kept some single images on ppts for talks and presentations.   However, we could not find one that would have matched the views selected for the other patients.  The image presented was the best we could find for this patient.  We apologise for this lack of uniformity and hope that the Reviewer appreciate we tried our best.

Question 5:   The MRI images provided illustrating Cases 2 and 4 show a clear involvement of the pons. How authors can be sure that the high grade glioma in those patients did not originate in the pons (a rather common site for high grade gliomas in children) secondarily invading the cerebellum? Please discuss this possibility and provide indication that involvement of the pons was not present in the remaining cases.

Response: As described in the text (the first paragraph under Results) in the original manuscript, these two tumors had their epicenters located in the cerebellar peduncles and were essentially cerebellar.    We also stated that the epicenters of all cases were in the cerebellum.   The cerebellar peduncles being white matter, it is conceivable that the pons could have been infiltrated but the latter was not the main epicenter.  They were not pontine tumors as other than their primary locations being the cerebellum, the tumors lacked the typical H3K27M mutation of diffuse infantile pontile glioma (DIPG), as described under the second last paragraph of Results.

Extra point : As suggested the journal, we made the manuscript 3,000 words or whereabout, so the actual manuscript is slightly different from previous version.  Thanks.

Reviewer 2 Report

Comments and Suggestions for Authors

This is a well written report of a series of five patients with High Grade Glioma of the cerebellum.  The authors have characterised these five tumors and described a rare entity with few other patients in the literature.  The report of the five patients adds to our understanding and the literature.  

Comments on the Quality of English Language

This is a well written manuscript describing a rare tumor in 5 children.  The authors have characterised the tumors and have added to our understanding about these tumors.

A few suggestions:

1. line 29: change "showed" to "shows"

2. line 65: change "5" to "Five"

3. line 67: change to "...,however, 15..." (add commas before and after however

4. line 79: change "While" to "Although"

5. line 186: change "etc" to words that describe the meaning the authors want to transmit

6. line 187: change "features" to "feature" 

Author Response

Comment: This is a well written report of a series of five patients with High Grade Glioma of the cerebellum.  The authors have characterised these five tumors and described a rare entity with few other patients in the literature.  The report of the five patients adds to our understanding and the literature. 

Response: Thank you for your helpful review and constructive comments.

Comment 1: line 29: change "showed" to "shows"

Response:  We changed "showed" to "shows” as suggested.  The change is highlighted.

Comment 2: line 65: change "5" to "Five"

Response:  We changed "5" to "Five."   The change is highlighted.

Comment 3:  line 67: change to "...,however, 15..." (add commas before and after however

Response:  We added commas before and after “however.” The change is highlighted.

Comment 4: line 79: change "While" to "Although"

Response:  We changed “While" to "Although."    The change is highlighted.

Comment 5: line 186: change "etc" to words that describe the meaning the authors want to transmit

Response:  We removed the word “etc”.   The change is highlighted.

Comment 6: line 187: change "features" to "feature"

Response:  We changed "features" to "feature" as recommended.  The change is highlighted.

Extra point : As suggested the journal, we made the manuscript 3,000 words or whereabout, so the actual manuscript is slightly different from previous version.  Thanks.

Reviewer 3 Report

Comments and Suggestions for Authors

After WHO 2016 of CNS tumor, many pediatric CNS tumor diagnosed by pathological findings are re-diagnosed to other entity by genetic pattern (ependymoma, HGG, etc…). After WHO2021, methylation pattern be highlighted to each tumor subtype. In this report, authors finally diagnosed five pediatric cerebellar HGG using multiple analysis methods. Using both genetic and methylation pattern is effective for pediatric rare CNS tumor.

My comment is as follow:

#1:2.1 patients

Reason for selection of your five patients is only institutes. Please add your selection in detail.

Author Response

Comments:  After WHO 2016 of CNS tumor, many pediatric CNS tumor diagnosed by pathological findings are re-diagnosed to other entity by genetic pattern (ependymoma, HGG, etc…). After WHO2021, methylation pattern be highlighted to each tumor subtype. In this report, authors finally diagnosed five pediatric cerebellar HGG using multiple analysis methods. Using both genetic and methylation pattern is effective for pediatric rare CNS tumor.

My comment is as follow:

#1:2.1 patients

Reason for selection of your five patients is only institutes. Please add your selection in detail.

Response: Thanks for the helpful review.

We retrieved cases from our archives for the periods 2002-2022 at the Chinese University of Hong Kong and 2013-2022 at Huashan Hospital, Shanghai.  The reason for the discrepancy was that Huashan Hospital, Shanghai only routinely admitted pediatric patients at a later period.  We retrieved from our system cerebellar high-grade gliomas and medulloblastomas to review and looked for potential cases, but did not look at all pediatric gliomas, as medulloblastoma is not a differential diagnosis for supratentorial lesions.  The cases described, including the three cases previously diagnosed as medulloblastomas, were the only cases of cerebellar high-grade gliomas we could identify eventually.  In the same period, there were 192 pediatric medulloblastomas excluding adult cases and consult cases at our institutions.    We outlined these under subsection “2.1 Patients” of “Materials and Methods.”

Extra point : As suggested the journal, we made the manuscript 3,000 words or whereabout, so the actual manuscript is slightly different from previous version.  Thanks.

Round 2

Reviewer 1 Report

Comments and Suggestions for Authors

After the authors response I still think that the the authors should include the MRI image of Case 4 in the main figure and not as a separate supplemental figures.

Author Response

Comment: After the authors response I still think that the the authors should include the MRI image of Case 4 in the main figure and not as a separate supplemental figures.

Response:  We now include the images of Case 4 in the main figure.